# POLICY GENERALIZATION IN CAPACITY-LIMITED REINFORCEMENT LEARNING

## ABSTRACT

Motivated by the study of generalization in biological intelligence, we examine reinforcement learning (RL) in settings where there are information-theoretic constraints placed on the learner's ability to represent a behavioral policy. We first show that the problem of optimizing expected utility within capacity-limited learning agents maps naturally to the mathematical field of rate-distortion (RD) theory. Applying the RD framework to the RL setting, we develop a new online RL algorithm, Capacity-Limited Actor-Critic, that learns a policy that optimizes a tradeoff between utility maximization and information processing costs. Using this algorithm in a 2D gridworld environment, we demonstrate two novel empirical results. First, at high information rates (high channel capacity), the algorithm achieves faster learning and discovers better policies compared to the standard tabular actor-critic algorithm. Second, we demonstrate that agents with capacity-limited policy representations avoid 'overfitting' and exhibit superior transfer to modified environments, compared to policies learned by agents with unlimited information processing resources. Our work provides a principled framework for the development of computationally rational RL agents.

## 1 INTRODUCTION

Generalization of learning is fundamental to intelligent behavior. For example, if a bird eats a poisonous butterfly, it must learn to avoid preying on that species again, even when no two butterflies look exactly alike. Similarly, in the game of chess a player must generalize intelligently from past experience as the same board position may never be encountered twice. The field of reinforcement learning (Sutton & Barto, 2018) provides a principled framework for the problem of learning, via trial and error, the long-term consequences of actions. However, by itself, RL lacks a formal theory of the generalization of learning. In practice, generalization is achieved through the combination of reinforcement learning algorithms with function approximation techniques, such as deep neural networks (DNNs) used to represent states, policies, and/or value functions. A separate thread of research has focused on learning hierarchical representations of policies, or learning extended action sequences ('options') that may enable stronger generalization and faster learning in complex tasks (Solway et al., 2014; Bacon et al., 2017). While these approaches have yielded notable success in numerous large-scale learning domains (e.g., Tesauro, 1995; Mnih et al., 2015; Silver et al., 2016), there is no extant body of theory that fully explains how and why such approaches can work well, and when they may fail in practice.

In the current paper, we instead seek to develop a principled mathematical basis for understanding generalization in the RL setting, building on theoretical cognitive science research on generalization in biological intelligence. In particular, within cognitive science the well-known 'Universal Law of Generalization' (Shepard, 1987) describes the strength of generalization from one stimulus to another, and has been demonstrated in multiple animal species and across a wide range of perceptual modalities. Recently, it has been shown that the Universal Law of Generalization emerges from any system that minimizes the expected cost of failing to discriminate between situations, subject to constraints on the ability to transmit or represent information (Sims, 2018). This theoretical result was obtained via the mathematical field of rate-distortion (RD) theory, which concerns the optimal solution to the problem of optimal, but 'lossy' data compression. According to this framework, biological organisms generalize because they must: they lack the information capacity to preserve all

state or environmental distinctions, and the resulting pattern of generalization depends, in a rational manner, on the expected cost of failing to discriminate.

Hence, RL and RD theory are complementary in their strengths and limitations. RL provides a theory of learning the long-term utility of actions via experience, but lacks a theory of generalization of learning. RD theory provides an elegant mathematical account of rational generalization in terms of information processing constraints, but assumes the long-term costs of equivocation between states or actions are known. While many readers may be familiar with RL or RD in isolation, few are likely to have more than passing familiarity with both. Hence, an important goal of the paper is a an accessible introduction of rate-distortion theory in the context of RL. Combining these two ideas, we consider the problem of *capacity-limited reinforcement learning*, in which the agent's learned mapping from states to actions is treated as a capacity-limited information channel that lacks the ability to represent behavioral policies with perfect fidelity. Such agents must learn, at best, a lossy compressed representation of an optimal behavioral policy. RD theory provides the theoretical bounds on what is achievable in this setting.

In pursuit of this idea, we develop a new online reinforcement learning algorithm, Capacity-Limited Actor-Critic (CL-AC) that learns a policy that optimizes a tradeoff between the expected utility of behavior, and the information rate of the agent's policy representation. Exploring the behavior of this algorithm in a 2D gridworld environment, we show that the introduction of capacity limits imparts two useful forms of generalization to RL agents. First, learned policies generalize across states during the course of learning in a principled manner. Consequently, at high information rates (high channel capacity), CL-AC achieves faster learning and higher accumulated reward compared to a standard tabular actor-critic (AC) algorithm. Second, we demonstrate that agents with capacity-limited policy representations exhibit superior generalization to modified environments, compared to policies learned by agents with unlimited information processing resources.

Our work is an instance of the general framework of computational rationality (Gershman et al., 2015), an emerging paradigm for understanding biological and artificial intelligence as efficient computing given information processing constraints. While information-theoretic channel capacity has long been understood as a relevant constraint on sensory coding in biological systems (Attneave, 1954; Barlow, 1961), as well as a model of economic decision-making in boundedly-rational humans (Sims, 2003) it has only more recently been explored as a constraint in RL (Grau-Moya et al., 2016; Ortega & Braun, 2011; Still & Precup, 2012; Rubin et al., 2012; Botvinick et al., 2015).

## 2    Reinforcement Learning and Rate-Distortion Theory

In this section we provide brief overviews of both computational RL and RD theory.

RL refers to a large family of different algorithms that address the problem of learning through experience how to select actions so as to maximize long-term expected reward. Here we briefly summarize the aspects relevant to the current work; a thorough introduction is given in (Sutton & Barto, 2018). The basic model for the interaction of an agent (artificial or biological) with a stochastic environment is a Markov decision problem (MDP), comprising a set of states ($\mathcal{S}$) that describe an environment, a set of candidate actions ($\mathcal{A}$) available to the agent in each state, and a dynamics function $p(s', r \mid s, a)$ specifying the joint distribution over subsequent state $s'$ and reward $r$, given a starting state $s$ and taking action $a$.

In the RL setting, the agent's goal is to learn (or approximate) an optimal policy function $\pi(a \mid s)$. In the present work, it will be useful to consider this policy function as an *information channel* that takes as input a current state, and produces as output a (stochastic) action to be followed in that state (e.g., $\mathcal{S} \rightarrow \mathcal{A}$). A fundamental construct in RL is an optimal value function, $V^\star(s)$, that indicates the expected long-term reward associated with being in a particular state $s$, and subsequently following an optimal behavioral policy. As shown below, an optimal value function is central to defining optimal performance for a capacity-limited policy channel.

For a physical agent with finite computational resources, it may be necessary or desirable to limit the amount of information stored about the policy or environment. Under such circumstances, what constitutes optimal performance? This question has an answer in the field of RD theory (Berger, 1971), which concerns the problem of optimal, but 'lossy' data compression. Within RD theory,

optimal performance for a communication channel is defined according to a given cost or loss function, $\mathcal{L}(x, y)$ that quantifies the cost of a signal $x \in X$ being transmitted as the value $y \in Y$.

More formally, the input to the channel is a sample from a source distribution $p(x)$, and the channel itself can be modeled as a conditional probability distribution $p(y \mid x)$. The goal for the channel is to minimize the cost of communication error specified by the loss function, $\mathcal{L}(x, y)$. For a given source and channel distribution, information theory provides a measure of the amount of information communicated (on average) by the channel in terms of its mutual information:

$$\mathrm{I}(X, Y) = \sum_y \sum_x p(y \mid x) p(x) \log \left( \frac{p(y \mid x) p(x)}{p(y) p(x)} \right). \tag{1}$$

Any physical communication channel is necessarily limited to transmitting information at a finite rate, and hence $\mathrm{I}(X, Y) \leq C$ for some finite value of $C$. Consequently, an optimally efficient communication channel is one that minimizes *channel distortion* (expected cost) subject to a constraint on information rate:

$$\textbf{Goal: } \text{Minimize } \mathrm{E}\left[\mathcal{L}(x, y)\right] \text{ w.r.t. } p(y \mid x)$$
$$\text{subject to } \mathrm{I}(X, Y) \leq C, \tag{2}$$

where the distortion $\mathrm{E}\left[\mathcal{L}(x, y)\right] = \sum_{x,y} \mathcal{L}(x, y) p(y \mid x) p(x)$.

A major topic within RD theory is finding efficient means of solving this constrained optimization problem. Blahut (1972) developed an efficient iterative algorithm for the case of channels with discrete input and output alphabets. This algorithm amounts to a minimization of the Lagrangian $\mathrm{I}(X, Y) + \beta \, \mathrm{E}\left[\mathcal{L}(x, y)\right]$ with respect to $p(y \mid x)$. The parameter $\beta > 0$ controls the tradeoff between the distortion and information rate of the channel, where higher values of $\beta$ correspond to channels with higher information rate and lower channel distortion.

Rather then being concerned with the mapping of abstract communication signals $x$ and $y$, we are instead interested in a channel that maps from states to actions, $\mathcal{S} \rightarrow \mathcal{A}$. In this case, a critical component in the application of RD theory to RL is the specification of an appropriate loss function for a capacity limited policy channel.

To that end, we introduce the *Bellman loss function*, $\mathcal{L}^\star(s, a)$, which defines the expected loss to an agent, relative to the optimal policy, associated with a policy channel producing as output an action $a$ in a given state $s$:

$$\mathcal{L}^\star(s, a) = V^\star(s) - \mathrm{E}[r(s, a) + \gamma V^*(s')]$$
$$= V^\star(s) - \sum_{s', r} p(s', r \mid s, a)(r + \gamma V^\star(s')). \tag{3}$$

Note that this is merely the difference in the optimal value for the starting state $s$, and the value of the immediate reinforcement plus the (possibly discounted) value of the resulting state $s'$, associated with the channel transmitting action $a$. Hence for all states and actions, $\mathcal{L}^\star(s, a) \geq 0$, with equality achieved for the optimal policy. Intuitively, the Bellman loss function states that when there is little or no difference between the cost of actions in a given state, there is no need to precisely represent a policy that distinguishes between them.

With a normative loss function defined, Algorithm 1 (Appendix) gives the complete Blahut algorithm adapted for computing an optimal, but capacity-limited policy channel. Note that this algorithm takes as input an optimal value function $V^\star$ for the MDP. In Section 3.2 we consider the problem of simultaneously learning an approximate value function, and using it to update an approximate Bellman loss function. However, we first assume a known optimal value function in order to examine in isolation the consequences of adopting a capacity-limited policy channel.

## 3 RESULTS

### 3.1 PERFORMANCE OF CAPACITY-LIMITED POLICIES

In this section, we evaluate the implications of adopting a capacity-limited policy in a 2D gridworld environment (Figure 1). The goal in this task is to navigate from a starting state to a terminal goal

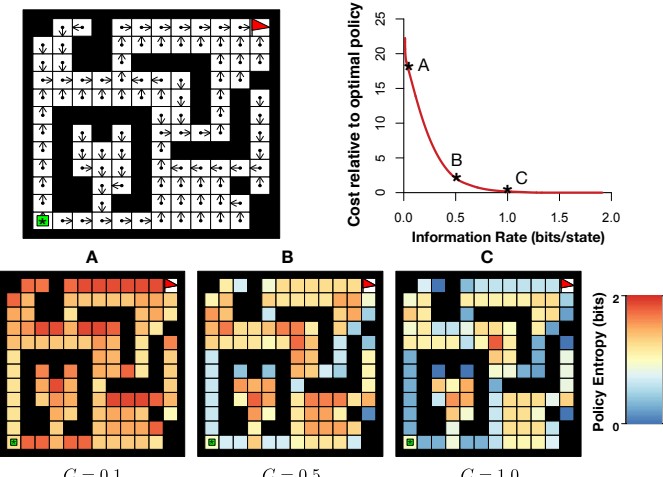

Figure 1: Top left: A gridworld environment where an agent must learn to navigate from a starting location (green asterisk) with the goal of reaching a target location indicated by the red flag. Crossing each open square has a reward of -1; colliding with walls (black squares) has a reward of -10. The optimal deterministic policy is shown for each state. Top right: Each point along the curve represents an optimal policy that achieves the maximal expected value at a given rate of information. Bottom: 3 policies illustrated at different points along the information rate distortion tradeoff curve. Colors in the plots illustrate the entropy of the policy in each state of the maze.

position. The states ($\mathcal{S}$), are defined by the current location of the agent within a 12x12 maze configuration. The actions ($\mathcal{A}$) are the possible directions that the agent can move (north, south, east, west) and the rewards are negative, such that the agent incurs a 1-point penalty per move and a 10-point penalty for colliding with walls.

The Bellman loss function defined in the previous section requires knowledge of the optimal value function. We start by assuming that $V^{\star}(s)$ is known, in the following section we relax this assumption. Note that the Blahut algorithm also requires a probability distribution over state occupancy, here we first assume a uniform distribution. For the small-scale gridworld of Figure 1, it is straightforward to solve for $V^{\star}$ using dynamic programming. The question explored in this section is the relative performance as capacity limits are varied on the policy function $\pi(a \mid s)$.

As an intuitive example, one can consider two kinds of behavioral policies for a gridworld environment. In one case, there are no information constraints, and the agent simply remembers the optimal action associated with each state. As the size of the state or action space grows, or for physical agents with limited computational resources, such a policy may be infeasible to store with perfect fidelity. In the case of information constraints, a 'compressed' policy might consist of a general plan (with high probability, move north or east) while storing more detailed instructions for key states where there are high costs for error.

To explore this idea, we apply the RD framework with the Bellman loss function (Algorithm 1) to obtain compressed policy functions at various levels of channel capacity. Figure 1, top right panel, illustrates the fundamental tradeoff between channel capacity and performance. In this gridworld environment specifying the optimal policy requires 2 bits per state (the maximum entropy with four possible actions). Because the optimal value function is known, each point along the curve represents the optimal achievable performance for a particular constraint on channel capacity. The results demonstrate that for the gridworld example considered, behavioral policies can be substantially compressed (by over half) without incurring significant cost to the agent.

The bottom row of Figure 1 illustrates three different capacity-limited policies ($C = 0.1, 0.5, 1.0$ bits per state), indicated by the labeled plot markers in the top right plot. The colors for each state indicate the entropy of the policy in that state. With very limited capacity the agent's policy is near-random. With increasing capacity, the agent focuses its limited computational resources on representing

important features of the environment with high fidelity (e.g. learning how to navigate key corridors), while using a stochastic policy in open areas of the maze. Notably, Figure 1 also illustrates that a form of exploration/exploitation tradeoff also emerges automatically from a capacity-limited policy: behavior is naturally more stochastic (exploratory) in states where the costs of error are unknown, or known to be small.

## 3.2  ON-LINE LEARNING WITH CAPACITY-LIMITED POLICIES

The results in the previous section illustrate the application of rate-distortion theory to a known and optimal value function. In this section, we consider the more difficult problem of simultaneously learning a value function online, and from it gradually improving a capacity-limited policy. Because of the recursive dynamics between an agent's current policy, future exploration, and updated policy, it is not *a priori* obvious that a capacity-limited agent will be able to discover or learn an effective policy.

We present in this section a detailed comparison between a standard RL algorithm, tabular Actor-Critic (AC) reinforcement learning Sutton & Barto (2018), and CL-AC, a novel extension that incorporates a tradeoff between the information rate of the policy and the loss in value incurred to the agent. The two algorithms are identical except for the addition of an information rate tradeoff parameter ($\beta$) introduced in CL-AC.

We chose AC as a basis for our extension for three reasons. First, this learning architecture distinguishes between a learned value function (the critic) and policy (the actor). In the present work, we limit our attention to exploring the impact of capacity limits on this latter component, although in principle the same approach could be extended to the value function as well. Second, standard actor-critic methods can directly learn a stochastic policy; this is a natural comparison to the stochastic (capacity-limited) policies acquired by the rate-distortion implementation. Lastly, the AC algorithm explicitly learns an approximate value function. The temporal difference (TD) error associated with this value function can be used as a training signal for updating the capacity-limited policy channel.

In particular, the previous section introduced the so-called Bellman loss function, $\mathcal{L}^{\star}(s, a)$. Computing this quantity required knowledge of the optimal value function for the task, $V^{\star}(s)$. The approach we adopt is simply to substitute an estimate of value function, $V(s)$, which is learned in an on-line manner. For the simple tabular case (discrete states, with no function approximation), the standard temporal difference (TD) error update rule for the state value function is

$$V(s) \leftarrow V(s) + \eta \left[ r + \gamma V(s') - V(s) \right]. \tag{4}$$

The parameter $\eta$ controls the learning rate, while $\gamma$ allows for temporal discounting (in our work $\gamma = 1$, implying no discounting). The estimated loss function $\mathcal{L}(s, a)$ is updated via the same temporal difference error. The required elements for learning are a starting state and action, $(s, a)$, the observed sample reward $r$ and resulting state $s'$, along with the current estimate of the value function. This yields:

$$\mathcal{L}(s, a) \leftarrow \mathcal{L}(s, a) + \eta \left[ V(s) - (r + \gamma V(s')) - \mathcal{L}(s, a) \right]. \tag{5}$$

Note that the first two terms inside the square brackets are online samples of the Bellman loss function (Eq. 3), substituting the estimated value function in place of the optimal value function. In the current work a common learning rate parameter is adopted for both the value and loss function.

With an estimated loss function, it is possible to obtain an optimal capacity-limited policy channel for that loss function using the Blahut algorithm. This typically involves an iterative algorithm that terminates only when it reaches a convergence criterion (as in Algorithm 1). However, as the value function itself is only a current estimate, we chose to terminate the Blahut algorithm after only a single iteration. Note that while this choice worked well in the current environment, it is not motivated by any underlying theory. In addition, the Blahut algorithm also requires a probability distribution over state occupancy, as a capacity-limited policy representation should be optimized with respect to the states that are most likely to be visited. In the current work we estimate the state distribution by keeping a tally of each time a state is visited. Note that this simple approach only approximates the true state distribution, as the agent's policy (and hence state distribution) changes over the course of learning. Theoretical convergence properties for this approach are not currently known. However, as we will show, the approach works well in the current setting. The complete CL-AC algorithm is presented in Algorithm 2 (Appendix). A more sophisticated approach could implement a trace decay

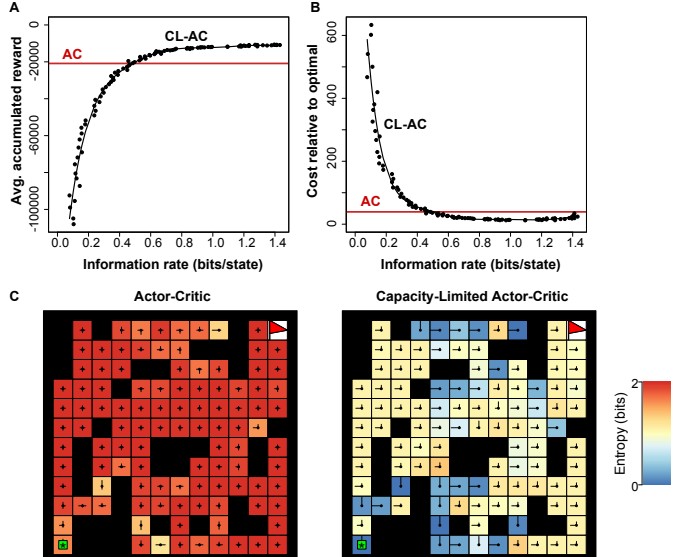

Figure 2: A) Average accumulated reward across 100 training episodes. Horizontal red line shows the performance of the standard (tabular) Actor-Critic algorithm. Black plot markers show the performance of CL-AC as a function of the information rate of the policy channel. B) Expected value of the policy learned at the end of 100 episodes, compared to the optimal policy for the task, as a function of the information rate of the policy. C) Comparison of the entropy of the policies learned by AC and CL-AC after one episode. Both algorithms were constrained to take the same sequence of actions and observed the same rewards. The learned policies are illustrated by line segments in each state that indicate the probability of taking each of the four actions (north, south, east, west).

associated with each state occupancy (Sutton & Barto, 2018). It is important to note that to minimize the possibility of divergence, the learned state distribution should change slowly relative to the policy, for example by using state eligibility traces with slow decay or decaying the state distributions once per episode rather than following every action selection.

We compared the performance of the standard AC and CL-AC algorithms on the gridworld environment introduced in Section 3.1. The CL-AC algorithm was tested at 100 different values of the capacity tradeoff parameter $\beta$. We note that there is likely no universally optimal setting for the parameter $\beta$, as it defines the tradeoff between loss of utility and the 'cost of information', and such a tradeoff is likely to be highly task-specific. For each parameter value, performance was averaged across 2,500 randomly generated maze environments. Each maze had the same start and terminal state, but had 36 randomly-placed walls, subject to the constraint that a viable path from the start to terminal state existed. The learning rate for both CL-AC and AC was fixed at $\eta = 0.1$, and no temporal discounting was assumed. The results are shown in Figure 2.

Notably, at high levels of $\beta$ (corresponding to higher channel capacity), the CL-AC algorithm outperformed its unlimited-capacity counterpart as evidenced by higher accumulated reward across the 100 training episodes (Fig. 2A), compared to standard AC RL. Figure 2B shows that at higher information rates, the final policy learned by CL-AC also achieved lower cost, relative to the optimal maze policy, compared to standard AC RL. Hence, these results show that it is possible, and even advantageous to simultaneously learn a value function in an on-line manner, and adapt a capacity-limited policy channel to the estimated value function. The superior performance of the CL-AC algorithm is due to the state generalization imparted by the capacity limited channel. When there is little knowledge about fine-grained distinctions in the environment (as is the case at the beginning of training), the Bellman loss function automatically encourages broad generalization of learning. This is illustrated in Figure 2C, where both the AC (left panel) and CL-AC (right panel) algorithms are trained for one episode, and are constrained to follow the same exact sequence of actions and observe the same rewards. Compared to AC, CL-AC exhibits far greater policy generalization given an equal amount of experience with the task.

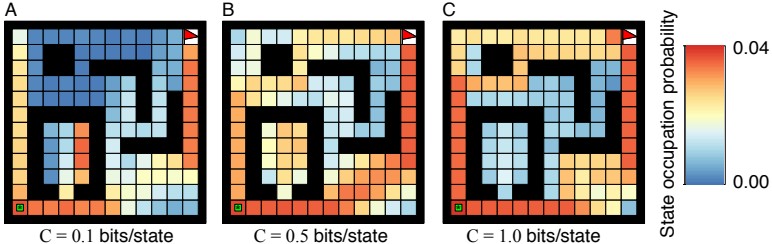

Figure 3: Average probability of occupying each state for learned policies using three different information rate constraints, averaged over 1,000 episodes.

To further illustrate the underlying behavior of the CL-AC policies and state representations, Figure 3 plots the state-occupancy distribution for different information trade-off rates. Figure 3A represents a highly capacity-constrained policy which is able to encode only rudimentary knowledge of how to navigate the maze (for example, move up or to the right with high probability). As a result, the agent becomes trapped in the upper right corner of the interior room with high probability. In contrast the policy in Figure 3B shows increasing sensitivity to the current state and differs across regions of the maze. For example, it's able to navigate through the corridor at the top left of the maze. Lastly, Figure 3C represents a policy with increasingly deterministic behavior. Note however, that it still strongly prefers to navigate along the bottom or left edges, rather than moving diagonally across the middle of the maze, even though all three paths take exactly 22 steps. The diagonal pathway requires a more complex (and hence higher information rate) policy in terms of an exact sequence of turns in order to avoid colliding with walls or becoming trapped. This behavior emerges directly from the combination of RD theory and the Bellman loss function, but is not seen in tabular AC.

### 3.3 GENERALIZATION TO NEW ENVIRONMENTS

In machine learning, it is commonly understood that complex models run the risk of *overfitting*: demonstrating good performance on a training set, but generalizing poorly to new environments. It has recently been shown that information-theoretic considerations for learning automatically favor simple models, independent of concerns about overfitting (Mattingly et al., 2018). This raises the question of whether standard RL agents suffer from overfitting in terms of their learned policies, and whether capacity-limited RL could naturally alleviate this problem. To test this idea, we trained RL agents in randomly constructed gridworlds for 100 episodes. At the end of training, we modified the maze by adding 8 additional walls placed in random locations, subject to the constraint that a viable path from the start state to terminal state existed. We thus test a modest form of generalization, focused on 'near' rather than 'far' transfer of learning.

We compared the performance of the standard AC algorithm, as well as the CL-AC algorithm using 100 different values of the capacity tradeoff parameter $\beta$. For each parameter value, performance was averaged across 2,500 randomly generated maze environments. Generalization performance was evaluated in terms of the expected value of the learned policy, as applied to the altered gridworld maze. The learning rate for both CL-AC and AC stayed fixed at $\eta = 0.1$.

Intuitively, one might think that capacity-limited agents should always be outperformed by agents with higher or unlimited capacity. Our results demonstrate that this is not the case. Figure 4 illustrates that generalization performance is highest at intermediate levels of channel capacity (approximately 0.5 bits per state). More surprising however, is the result that CL-AC at intermediate capacity levels shows superior generalization performance compared to the standard AC algorithm. An intuitive account is that compared to the standard AC algorithm, capacity limits force agents to concentrate representational resources on the most critical states. This can often have the effect of increasing stochasticity in states where there are low costs for error, and hence naturally encourage exploratory behavior. However, an added capacity constraint in the RD framework is conceptually different from simply increasing stochasticity in behavior. For example, note that a policy that deterministically follows the same action regardless of the current state also has an information rate of zero. The policy learned via CL-AC represents the (approximate) optimal achievement of utility within an information constraint, and this policy can exhibit both increases and decreases in stochasticity in

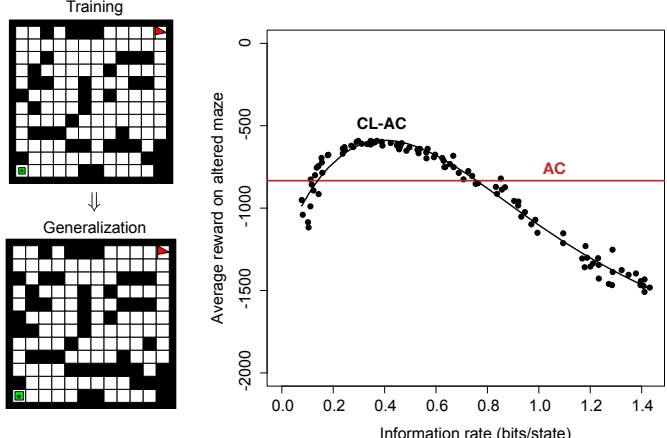

Figure 4: Left: Agents were trained on a randomly constructed maze, and evaluated on a maze with additional walls randomly placed. Right: Average generalization performance for CL-AC, as a function of the trade-off parameter $\beta$. Generalization performance of the standard AC algorithm is shown by the horizontal red line.

different states. We show, empirically, that such policies generalize well to modest modifications of the trained environment.

These results reiterate the benefits of the state generalization imparted by capacity-limited policies, as they constrain the system towards more robust policies that avoid issues associated with overfitting in policy space. This continues to be a central topic of interest within the machine learning and cognitive science literature, as a hallmark of both biological and artificial intelligent systems is the ability to flexibly abstract, summarize and generalize learning (Botvinick et al., 2015; Sims, 2018).

## 4 CONCLUSIONS

This paper describes the application of rate-distortion theory to the learning of efficient (capacity-limited) policy representations in the reinforcement learning setting. The current work is related to several other papers that have explored the problem of information constraints in MDPs. The approaches of (Rubin et al., 2012) and (Larsson et al., 2017) optimized a free energy formulation that is similar to the rate-distortion objective developed here, but did not consider online learning via RL. Van Dijk & Polani (2013) demonstrated using a similar grid-world task that imposed informational constraints could lead to intrinsically organized behavior, such as the emergence of sub-goals. Larsson et al. (2017) also explored policies that are not state-specific and addressed generalization as a series of state space abstractions. In (Still & Precup, 2012), an online learning algorithm was developed that derived from Q-learning, but it required the availability of a model of the environment. Further, this work did not directly compare learning or generalization performance against alternative approaches.

While building on these related approaches, the current paper accomplishes several goals. We introduce the *Bellman loss function*, which provides the normative cost function for a capacity-limited policy channel. Using this construct, we introduce a novel model-free RL algorithm, CL-AC, that learns a policy that optimizes a tradeoff between utility and information rate. We show that this algorithm has good learning performance in a typical gridworld environment, although future work must explore its convergence properties in more complex domains. Compared to the standard AC algorithm, CL-AC has several advantageous properties. At high information rates, CL-AC achieves superior performance, in terms of faster learning and policies that achieve higher expected reward. In addition, we demonstrate that at intermediate capacities, compressed policies exhibit greater robustness and generalization to novel environments. Notably, While there exists no general rule for choosing the appropriate tradeoff rate . Through its focus on the optimal minimization of cost subject to information resource constraints, RD theory provides a natural framework for the development and specification of computationally rational learning agents.

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

APPENDIX

---

**Algorithm 1:** Blahut algorithm adapted for computing a capacity-limited policy channel

---

Input: an optimal state value function $V^\star(s)$
Input: reward & state transition distribution $p(r, s' \mid s, a)$
Input: a state occupancy distribution $p(s)$
Parameters: capacity tradeoff $\beta > 0$
Initialize $\pi(a \mid s)$ arbitrarily
**for** *each action $a$* **do**
    **for** *each state $s$* **do**
        $\mathcal{L}(s, a) \leftarrow V^\star(s) - \sum_{s', r} p(s', r \mid s, a)\,[r + \gamma V^\star(s')]$
    **end**
**end**
**repeat**
    **for** *each action $a$* **do**
        $q(a) \leftarrow \sum_s \pi(a \mid s)\, p(s)$
    **end**
    **for** *each state $s$* **do**
        **for** *each action $a$* **do**
            $\pi(a \mid s) \leftarrow \frac{q(a)\exp(-\beta\mathcal{L}(s,a))}{\sum_{a_j} q(a_j)\exp(-\beta\mathcal{L}(s,a_j))}$
        **end**
    **end**
**until** *until convergence*;
**return** $\pi(a \mid s)$, a capacity-limited policy channel

---

**Algorithm 2:** Capacity-Limited Actor-Critic for episodic tasks

---

Input: an initial policy channel $\pi(a \mid s)$
Input: an initial state value function $V(s)$
Input: an initial state occupancy count $z(s) \geq 0$
Parameters: learning rate $0 < \eta \leq 1$; discount $0 \leq \gamma \leq 1$; capacity tradeoff $\beta > 0$
**repeat**
    set $s$ to initial state
    **while** *$s$ is not terminal* **do**
        $z(s) \leftarrow z(s) + 1$
        **for** *each state $s$* **do**
            $p(s) \leftarrow \frac{z(s)}{\sum_{s_j} z(s_j)}$
        **end**
        $a \sim \pi(\cdot \mid s)$
        Take action $a$, observe $s'$, $r$
        $V(s) \leftarrow V(s) + \eta\,[r + \gamma V(s') - V(s)]$
        $\mathcal{L}(s, a) \leftarrow \mathcal{L}(s, a) + \eta\,[V(s) - (r + \gamma V(s')) - \mathcal{L}(s, a)]$
        **for** *each action $a$* **do**
            $q(a) \leftarrow \sum_s \pi(a \mid s)\, p(s)$
        **end**
        **for** *each state $s$* **do**
            **for** *each action $a$* **do**
                $\pi(a \mid s) \leftarrow \frac{q(a)\exp(-\beta\mathcal{L}(s,a))}{\sum_{a_j} q(a_j)\exp(-\beta\mathcal{L}(s,a_j))}$
            **end**
        **end**
        $s \leftarrow s'$
    **end**
**until** *finished*;

---

