# OpenReview forum: "Policy Generalization In Capacity-Limited Reinforcement Learning"
_ICLR.cc/2019/Conference_

### Official Review · AnonReviewer2 · 2018-10-14
**algorithm not guaranteed to work**

**Rating:** 5
**Confidence:** 4

**Review:**

Disclosure: I reviewed this paper for a different conference but have read the new manuscript and noted the changes.

Summary:
The paper considers a very novel (but important) RL context where the agent has a constrained amount of information for representing a policy.  The authors use techniques from rate-distortion theory to generate a clever Bellman loss function that can be used (1) in a context where V*(s) is already known, and more importantly (2) with an actor-critic architecture (CL-AC) where the value function is being learned online.  CL-AC is shown to actually achieve higher converged and cumulative rewards than AC in many grid world domains and is shown to be advantageous in a transfer learning setting as well.

Review:

The ideas in the paper are very well described and laid out.  The experiments are on grid worlds but for such a novel problem like this I think they are at the right level because they allow the reader to understand the results.  The empirical results are compelling, but I have a strong technical concern about the convergence issue noted by the authors (which was also communicated to the authors in a previous conference’s review session).

My main concern is, as the authors noted, the required state occupation probability p(s) for RDT is approximated in a way that could lead to bad behavior in the RL algorithm.  What we’re seeing here is the application of an RDT procedure that was designed for a static distribution being applied to a dynamic distribution of states (that can change based on the policy).  In RL, there is no guarantee that the previous occupation probabilities have anything to do with the current policy’s induced distribution.  In a hallway world with a decent reward down the left and a bigger reward to the right, an algorithm might start off by going down the left side several times, making the probabilities of states on the right 0.  If I am reading the algorithm right, the states on the right are going to be essentially dismissed as unlikely, and the “go right” action (which is optimal) will likely be compressed out, since the states it should be used in are considered unlikely.  More succinctly, early trajectories will bias p(s) and cause the algorithm to essentially want to optimize the policy for that distribution, likely causing it to stay in that distribution.  Even more dangerously, there may be cases where this could cause the algorithm to thrash between policies as p(s) oscillates between different parts of the state space.

In order to improve this paper and make it suitable for publication, the authors should at least empirically demonstrate how different state occupation probability approximations affect the algorithm.  A good example is the trace-decay probabilities mentioned (but not implemented) in the paper.  If the paper compared that approach to the current approach, and showed an environment where one or both approaches failed to act correctly, that would complete the scientific result. Right now, only one approximation is demonstrated, and as detailed above, its behavior is suspect.

While most of the empirical results are well explained, the behavior in Figure 2B, where CL-AC is outperforming standard AC remain unclear. I understand that in 2A (avg. cumulative reward), CL-AC may be inducing a more efficient exploration policy and therefore the rewards during learning will be better.  But in 2B, we are just looking at the final policy.  Was standard AC not able to find the optimal policy after 100 episodes?

The results in the transfer learning context (Figure 3) are well done and produce a very interesting curve.

Reference 9 appears to only be available as an arXiv pre-print.  Papers that have not been properly vetted by peer review should not be cited in an ICLR paper unless they are extremely necessary, which this does not appear to be.


Typo: Page 6 – sate -> state

---

> ### Author Response · Authors · 2018-11-27
> **Response to Reviewer 3:**
>
>
> Thank you for your thorough review. We greatly appreciate your time and feedback on this work. We agree that the dynamic nature of learning implies that the simple tally approach explored here does not accurately represent the state occupancy distribution of the existing policy. We are currently exploring the implications of this fact, and applying an approach that uses a trace decay associated with each state. We recognize your concern and the possibility that divergence may occur between the learned policy and state distribution. To minimize this risk, we suggest that the approximated state distribution should change slowly relative to the policy, for example by using state eligibility traces with a slow decay rate or decaying the state distributions once per episode rather than once per every action selection. A discussion of these issues has been added to the revised version of the paper.
>
> Although these methods offer a great deal of efficiency and flexibility, we agree that at present we can offer no guarantees of convergence, particularly when extending the current algorithm to large state/action spaces. Notably however, most guarantees for convergence in RL in general apply only to very limited circumstances, such as the tabular-based cases of TD algorithm methods (Sutton & Barto, 2018, p. 196). As cautioned by Sutton & Barto (2018) convergence using stochastic approximation methods is not guaranteed unless restrictive conditions are met, such as infinite exploration or sufficiently small learning rates. Most “real world” algorithms based on Deep-RL also lack convergence guarantees, especially when off-policy, nonlinear function approximation, and bootstrapping methods are combined in one RL algorithm, leading to the “deadly triad” issue (Sutton & Barto, 2018,  p. 249). Of course, the theoretical possibility of divergence is less concerning in the face of empirical success. As we are developing a relatively novel approach within RL, we believe the first step is to demonstrate the algorithm's behavior in small-scale setting. Our future work seeks to extend this work to more complex learning domains.

---

### Official Review · AnonReviewer3 · 2018-11-02
**Nice introduction/summary to rate-distortion in RL with illustrative examples**

**Rating:** 7
**Confidence:** 4

**Review:**

(score raised from 6 to 7 after the rebuttal)
The paper explores the application of the rate-distortion framework to policy learning in the reinforcement learning setting. In particular, a policy that maps from states to actions is considered an information theoretic channel of limited capacity. This viewpoint provides an interesting angle which allows modeling/learning of (computationally) bounded-rational policies. While capacity-limitation might intuitively seem to be a disadvantage, intriguing arguments (based on solid theoretical foundations, rooted in first principles) can be made in favor of capacity-limited systems. Two of the main-arguments are that capacity-limited policies should be faster to learn and be more robust, i.e. generalize better. After thoroughly introducing these arguments on a less formal level and putting them into perspective with regard to reinforcement learning and related work in the literature, the paper demonstrates these properties in a toy grid-world example. When compared against a vanilla actor-critic (AC) algorithm, the capacity-limited version is shown to converge faster and reach better final policies. The paper then extends the basic version of the algorithm, which requires knowledge of the optimal value function, towards simultaneously learning the value function. While any theoretical guarantees are lost, the empirical results are still in line with the theoretical benefits, outperforming vanilla AC and producing better results in previously unencountered variations of the grid-world environment.

The paper is very well written and the toy-examples illustrate the theoretical advantages in a very nice and intuitively understandable way. The topic of modeling capacity-limited RL agents and exploring how capacity-limitation is an advantage, rather than a “bug” is very timely and important. In particular, rate-distortion theory might provide key-insights into building agents that generalize well, which is among the major open problems in reinforcement learning. The paper is thus very timely and highly relevant to a broad audience.

The main weakness of the paper is that it is of quite limited novelty and that the brute-force approach towards using Blahut-Arimoto in RL is unlikely to scale to large, complex state-/action-spaces without major additional work. Continuous state-/action-spaces are in principle covered by the theory, but they come with additional caveats and subtleties (I appreciate the authors using discrete notation with sums instead of integrals). Additionally, when simultaneously learning the value function (in the online setting), any guarantees about Blahut-Arimoto convergence are lost. However, solving either of these issues is hard and many attempts have been made in the communications community. Despite these weaknesses I argue for accepting and presenting the paper at the conference for the following reasons:
- modelling capacity-limited agents via ideas from rate-distortion theory (which is very closely related to free-energy optimization, such as ELBO maximization, Bayesian inference and the MDL principle) is an underrated topic in reinforcement learning. On a conceptual level, the strong idea is that moving away from strict optimization and infinite-capacity systems is not a shortcoming but can actually help building agents that perform better and generalize better. This is not a well established idea in the community. The paper does a good job at introducing the general idea, illustrating it intuitively with toy examples and pointing out relevant literature.
- Simultaneously learning the value function is necessary in the RL setting, but breaks quite a bit of the theory. However, very similar ideas seem to work quite well empirically in other settings, such as for instance ELBO maximization in VAEs, where the “value function” is the log-likelihood (under the decoder), which is learned simultaneously while learning a “policy” (the encoder) under capacity limitation (the KL term). Similar arguments can be made for modern InfoBottleneck-style objectives in deep learning. Based on this empirical observation, it is not unlikely that simultaneous learning of the value function works reasonably well without catastrophically collapsing in other settings and tasks.
- While achieving a solution that strictly lies on the rate-distortion curve might be crucial in communications, it might be of lesser significance for building RL agents that generalize well - slight sub-optimalities (solutions that lie off the RD curve) should still yield interesting agents. Therefore, losing theoretical guarantees might be less severe for simply exploring how much the idea can be scaled up empirically.

Minor issues:
1) While the paper, strictly speaking, introduces a novel algorithm and the Bellman loss function (which requires knowledge of the optimal value function), I think that the main contribution is a clear and well-focused introduction of rate-distortion theory in the context of RL, including very illustrative toy examples. I do consider this an important contribution.

2) Transfer to novel environments. The final example (Fig. 4) does show that the capacity limited agent performs better in novel environments. However, I’m not entirely convinced that this demonstrates “superior transfer to novel environments” (from the abstract). While the latter might very well arise from capacity-limitations, I think that in the example in the paper there is not too much transfer going on, but the capacity-limited agent simply has a more stochastic policy which helps if unknown walls are in the way. After all, the average accumulated reward of the capacity limited agent does also decrease significantly in the novel environment - it simply does a slightly better random walk than the AC (correct me if I’m wrong, of course). On page 7, last paragraph this is phrased as: “agents retain knowledge of exploratory actions”. In my opinion this wording is a bit too strong to simply describe increased stochasticity.

3) Since the paper does provide a good overview over the literature, I think it would help to mention that the current main approach towards generalizing (deep) RL is via hierarchical RL (options framework, etc) and provide a good reference.

4) At the very end of the intro you might also want to mention that rate-distortion has been used before in the context of decision-making (not RL), for instance under the term rational inattention.

5) Page 5, last paragraph: the paper mentions that one Blahut-Arimoto iteration is enough. This is an empirical observation, justified by the toy experiments. However, the wording sounds like this is a generally known fact. Please rephrase to emphasize that this must not necessarily hold true in general and that convergence behavior might crucially depend on this.

6) It would be good to give readers some guidance towards choosing beta if doing an exhaustive grid-search is infeasible. I am aware that there is no good general rule or recipe, but perhaps something can be added to the discussion (even if it is just mentioning that there is no good heuristic, etc. - however, there should be plenty of research in communications that deals with estimating the RD curve from as few points as possible).

7) Please consider adding this reference - it has a very similar objective function (but for navigating towards multiple goals) and is very much in line with some of the theoretical arguments.
Informational Constraints-Driven Organization in Goal-Directed Behavior - Van Dijk, Polani, 2013.

---

> ### Author Response · Authors · 2018-11-27
> **Response to Reviewer 2:**
>
>
> Thank you for your thoughtful and thorough review. Your review identified a number of strengths and weaknesses, but concluded that the former outweigh the latter, and that the work is "very timely and highly relevant to a broad audience". We strongly agree with this conclusion.
>
> More specifically, in your review, you identified two main weaknesses of the current paper: "[A] it is of quite limited novelty and [B] that the brute-force approach towards using Blahut-Arimoto in RL is unlikely to scale to large, complex state-/action-spaces without major additional work".
>
> Regarding [A], we respectfully disagree that the paper is of limited novelty. Although there are a small handful of papers that have previously explored information-theoretic limits in RL (cited in our manuscript), this topic is, comparatively speaking, unexplored terrain. Regarding the specifics of the novelty of our approach, we feel confident that our work is the first to propose an online, model-free, capacity-limited RL algorithm, and empirically demonstrate its performance against standard (non-capacity-limited) approaches.
>
> Regarding [B], we agree that substantial additional work would be needed to extend the current algorithm to large state/action spaces. The scope of the work involved prevents us from incorporating this into the manuscript within the author response period. However, this should not imply that the approach cannot be scaled up.
>
> Our demonstration of the algorithm utilized a tabular representation of policy and value functions. Indeed, part of the value of this approach is to demonstrate that a tabular state representation does not preclude rational generalization. But there is nothing intrinsic to RD, RL, or their combination, that requires a tabular representation. Indeed, any standard function approximation scheme could be substituted into the current algorithm, from tile coding to deep NNs. We envision a policy represented as a neural network, with an information rate constraint placed on the training of the network. In other words, the network is trained to approximate an optimal capacity-limited channel (in the rate-distortion sense). In standard practice, overfitting is prevented (if it is considered at all) via careful design of the network architecture. In contrast, rate-distortion theory adds principled regularization that works regardless of the network architcture. While implementing and fully testing this idea would require substantial additional work, we feel that such an effort would be a major separate contribution, and does not diminish the value of the current work.
>
> Your review also identified a number of minor issues, here we briefly summarize how the manuscript has been revised in response to each of these points.
>
> 1) Thank you for pointing this out. We have made minor revisions to the manuscript to underscore this aspect of the paper's contribution.
> 2) We agree that the demonstration of transfer to modified maze environments is a fairly limited type of generalization. We have moderated some of the language used to describe this type of "generalization". In effect it is a type of regularization on the agent's policy. However, it is not quite the same as enforcing a more stochastic policy. Capacity limits can actually *increase* the determinism of behavior (for example, a policy that takes the same deterministic action regardless of the state has an information rate of zero). The actual balance of determinism and stochasticity learned by the agent represents the (approximate) optimal achievement of utility with an information constraint.
> 3) Thank you for the suggestion, we have added a reference to point to related work in hierarchical RL.
> 4) We have added a reference to the closely related "rational inattention" framework.
> 5) Thank you, we clarify that using one Blahut iteration is not motivated by any strong theoretical basis.
> 6) We added a brief discussion regarding the choice of the beta parameter to address this.
> 7) Thank you for pointing out this citation, which we were previously unaware of. We have incorporated it into the manuscript.

---

> > ### Comment · AnonReviewer3 · 2018-11-29
> > **Response to Rebuttal**
> >
> > Thank you for the detailed responses and the updated the manuscript. Let me add the following:
> >
> > Regarding [A] (novelty): perhaps my initial phrasing was a bit harsh and I apologise for that. I do agree that the paper combines elements/ideas from RD and RL in an original fashion. Let me try to clarify: for researchers that are familiar with RD in the context of decision-making/RL, I think the level of "surprise" is not too high, but they might very well appreciate the demonstration of some of the appealing aspects of the theory in clear and simple examples (and the clarity of the writing). Perhaps more importantly, I acknowledge that ideas from RD are not very well known in the RL community and I personally think that the paper does a very good job in making these ideas accessible to a broad RL audience - how to judge that aspect in terms of "objective novelty" is unclear to me (in the same way that a review paper is strictly speaking never "novel").
> >
> > Regarding [B] (scaling-up): I agree with the authors that candidate-approaches for empirically scaling up the approach are quite clear (and that they are on the scale of a separate publication). I also agree that RD offers the theoretical advantage of very principled regularization, which promises to lead to better generalization. Whether this theoretical advantage and others will carry over to specific implementations that scale well, remains an open question. I personally think it is worthwhile giving it a serious shot (also given the empirical observation that very similar ideas work quite well in the context of training generative models and auto-encoders). However, one aspect that I find non-trivial and that might easily break naive scaling-up is that Blahut-Arimoto requires a marginalization step (computing the "aggregate posterior" which is also the "optimal prior"). Doing this in a sample-based fashion in high-dimensional problems might turn out to be very difficult (or require very large sample-size), especially if certain parts of the state-space are never/rarely visited. It is currently unclear to me how brittle convergence behaviour is with respect to errors in this step, but I could easily imagine running into the problem of "mode collapse" or unnecessarily greedy policies with degenerate exploration.
> >
> > Regarding 2): I agree with the authors' response, particularly that one appealing aspect of RD is that it "automatically" leads to policies that know when to be deterministic and when to be noisy. However, I think that the example shown in the paper is perhaps not complex enough to allow for complex generalization behavior, which is why I appreciate that the authors have toned down the language a bit.
> >
> > I have chosen to raise my score, but that is not to say that I disagree with Reviewer 2. I want to strongly encourage the authors to take the issues raised by Reviewer 2 into account. I personally, think it is interesting to see how far the method can be naively scaled up before problems start emerging. But it will also be necessary to advance the  understanding of how to scale up in a theoretically sound way. I think that both aspects can be worked on in parallel to some degree, but it should be kept in mind that the empirical results currently rest on a very brittle theoretical foundation. I also want to encourage the authors to be very clear and upfront about the current shortcomings and the current state of this line of research.

---

### Official Review · AnonReviewer1 · 2018-11-03
**Inciteful and of general interest.**

**Rating:** 7
**Confidence:** 3

**Review:**

## Summary

The authors identify a synergy between the rate distortion (RD) and reinforcement learning (RL) literature. RD work shows how to optimise resources when capacity is limited and the authors transfer this idea to RL and posit a novel algorithm based on the Actor critic algorithm. In experiments this is shown to learn more quickly and transfer between similar tasks more easiliy than the conventional AC algorithm.

This is a genuinely inciteful piece of work and may be of very significant interest to the community. Particuarly to those in transfer learning, heirarchical learning and other areas of RL where an adaptable rate limited policy is an advantage.

The experiments are limited to a single domain, and ideally this would be demonstrated across more than just those examples explored. However, I think that the value of the theoretical advance, and the clarity/readability of the paper warrants acceptance.

---

> ### Author Response · Authors · 2018-11-27
> **Response to Reviewer 1:**
>
>
> Thank you for your review. We agree that much of the strength of the current paper lies in introducing a relatively novel, and principled theoretical framework to a broad audience. We view this work as a jumping off point for readers who have some background in rate-distortion theory, or reinforcement learning. However, few in the machine learning community have more than passing knowledge of both. We feel the demonstration of the approach in a small-scale environment allows for an unhurried exposition of the theory and demonstration of its behavior. Many of the details of the algorithm can be substituted, for example utilizing deep-NNs for approximating an optimal-but-capacity-limited channel, enabling the extension to large or continuous state-spaces. Given the recommended page limits of the ICLR proceedings, we opted for a fuller and more accessible exposition of the theoretical approach, rather than an extensive demonstration of its empirical performance.

---

### Meta-Review · Area_Chair1 · 2018-12-14
**Interesting and novel work, but with a severe theoretical flaw**

**Confidence:** 4
**Recommendation:** Reject

**Metareview:**

The paper studies RL from a rate-distortion (RD) theory perspective.  A new actor-critic algorithm is developed and evaluated on a series of 2D grid worlds.

The paper has some novel idea, and the connection of RL to RD is quite new.  This seems like an interesting direction that is worth further investigation.  On the other hand, all reviewers agreed there is a severe flaw in this work, casting a doubt where RD can be directly applied to an RL setting because the distribution is not fixed (unlike in standard RD).  This issue could have been addressed empirically, by running controlled experiments, something the the paper might include in a future version.